# Comparison of machine learning methods for genomic prediction of selected *Arabidopsis thaliana* traits

**Ciaran Michael Kelly**[¤], **Russell Lewis McLaughlin** *

Smurfit Institute of Genetics, Trinity College Dublin, Dublin, Ireland

¤ Current address: School of Pharmacy and Biomolecular Sciences, RCSI University of Medicine and Health Sciences, Dublin, Ireland
* mclaugr@tcd.ie

## Abstract

We present a comparison of machine learning methods for the prediction of four quantitative traits in *Arabidopsis thaliana*. High prediction accuracies were achieved on individuals grown under standardized laboratory conditions from the *1001 Arabidopsis Genomes Project*. An existing body of evidence suggests that linear models may be impeded by their inability to make use of non-additive effects to explain phenotypic variation at the population level. The results presented here use a nested cross-validation approach to confirm that some machine learning methods have the ability to statistically outperform linear prediction models, with the optimal model dependent on availability of training data and genetic architecture of the trait in question. Linear models were competitive in their performance as per previous work, though the neural network class of predictors was observed to be the most accurate and robust for traits with high heritability. The extent to which non-linear models exploit interaction effects will require further investigation of the causal pathways that lay behind their predictions. Future work utilizing more traits and larger sample sizes, combined with an improved understanding of their respective genetic architectures, may lead to improvements in prediction accuracy.

## Introduction

### Genomic prediction

A major goal in the field of genetics is to make use of genotypic information to predict phenotypic values at both the individual and population level. When considering a population, the amount of variance that can be explained by genetics is bounded by the broad-sense heritability ($H^2$). This parameter, the ratio of genotypic to phenotypic variance, is not fixed in time or space, and can vary substantially between traits [1].

In practice, genomic prediction has been achieved largely through the development of linear models, often with great success [2, 3]. However, linear models are naturally limited by their inability to account for all non-linear statistical effects. They are instead bounded by the

**Data Availability Statement:** All genotypic and phenotypic data is available from the 1001 Arabidopsis Project (https://1001genomes.org/data-center.html) All optimizing and statistical code

is available from Github (https://github.com/ciaranoceallaigh96/arabidopsis_ml/).

**Funding:** This study was supported by Science Foundation Ireland (17/CDA/4737). Russell McLaughlin also receives support from the Motor Neurone Disease Association (879-791) and Science Foundation Ireland (16/RC/3948). The funders had no role in study design, data collection and analysis, decision to publish, or preparation of the manuscript.

**Competing interests:** The authors have declared that no competing interests exist.

narrow-sense heritability ($h^2$) which is defined as the proportion of additive genetic variance to overall phenotypic variance. Thus, linear models may not account for all of the total genetic variation, such as that which could arise from complex non-linear interactions between genetic variants across the genome, i.e. statistical epistasis [4].

It must be noted that linear approaches do not correspond to only modelling additive gene action at the molecular level. Importantly, non-additive biological effects can be partly subsumed into additive variance and indeed be exploited for prediction purposes [1]. It is for this reason that many negate the importance of non-linear models for improving performance or accounting for "missing heritability", that is the difference that exists between the phenotypic variance estimated to be explainable by genetic variation and the current performance of the models that have thus far been developed [5–7].

Nevertheless, a substantial amount of variation may not be captured by additive variance and it is not necessarily the case that the standard parameterization of variance into additive and non-additive effects is always the ideal framework for approaching a given genomic prediction task [8–10].

The optimal prediction method depends on the actual genetic architecture of the trait, even when the magnitude of non-additive variance is significantly less than the additive variance. It cannot be assumed *a priori* that linear models will be the most favourable prediction approach for any given trait of interest. Furthermore, moving beyond accounting for only the narrow-sense heritability of a typical complex trait will require more non-linear modeling of interaction effects.

Notwithstanding these limitations, linear prediction models are generally simple to build, interpret, and transfer across datasets. As stated above, they can indeed capture some non-linear genetic effects and they have seen much success over the last few decades in predicting individual plant phenotypes and in genomic selection using single nucleotide polymorphism (SNP) data [3, 11]. For this reason they remain an important benchmark against which to gauge any other predictive model. The linear approach can involve measuring the associations between genetic markers and the trait of interest and, simultaneously or subsequently, generating a polygenic prediction model from the subsets of those highlighted quantitative trait loci (QTLs) that are most associated. Common association and prediction methods include genome-wide association studies (GWAS) and genomic best linear unbiased prediction (gBLUP).

The gBLUP method makes use of a linear mixed model, where SNPs are modelled as random effects, to evaluate the combined effects of all QTLs in the genome, with the assumption that effect sizes are small and normally distributed. gBLUP was introduced as an extension of the traditional best linear unbiased prediction (BLUP) through the incorporation of a genomic relationship matrix (GRM) rather than the previous reliance on pedigree information in accounting for covariance between relatives in the population [12, 13]. The GRM consists of the realized proportion of shared genome for each pair of individuals within the matrix. gBLUP is generally more accurate than BLUP as it relies on the actual relatedness between individuals (even distantly so) rather than their assumed relationship. Any new individual with known genotype and unknown phenotype can get a reasonable estimate for the trait value of interest using this method. It is a standard method of choice in animal and plant prediction tasks, while also being applicable to human data [14].

Recently, the field of machine learning has come to the fore in offering promising methods for achieving higher polygenic prediction accuracies, as they have greater flexibility to model the complexity that may underlie the genetic proportion of phenotypic variance. It is possible that some machine learning methods can make progress in accounting for some of the missing heritability that remains to be explained.

## Machine learning

Machine learning refers to a set of algorithms that are trained to detect and exploit complex patterns in data that can then be used as predictors on separate unseen test datasets. For genomic prediction, the training dataset generally consists of individual-level genotype inputs and phenotype value outputs. As the training individuals' trait values are labelled, this is a supervised class of prediction problem [15].

Although more simple models may provide causal insights into how input variables relate to the output (i.e. genotype to phenotype), the appeal of some machine learning families lies in their greater flexibility to model complexity, such as that which may arise from the genetic architecture of a trait.

In doing so, it is important to note that not all machine learning methods are non-linear, such as regularised regression models (Ridge and LASSO) or linear support vector machines (SVMs). However, these approaches may, in some instances, still be more suitable for the genetic architecture underlying certain traits (e.g. via effective regularisation and feature selection) and the resulting models may provide a more appropriate framework to capture some non-additive biological action in the marginal effects of risk alleles.

When training a machine learning model, many hyperparameters of the chosen model type may need to be optimized, depending on the specific machine learning method in question. This can be achieved through searching various combinations of these hyperparameters, with time and computational power usually being limiting factors. As a result of these complex optimization techniques, the final models are often not readily interpretable because they can involve complicated high-dimensional non-linearities.

Several previous studies have been published on the use of machine learning for genomic prediction in plants [16–19]. In general, some machine learning methods are reported to outperform linear models, although not universally so, and the magnitude of that benefit is variable [20]. In an *Arabidopsis thaliana* context, the work of Farooq et al. using the RegMap dataset has shown that non-linear models can out-compete traditional models for some complex traits [21, 22]. Raimondi et al. have found that neural networks are useful for genomic prediction using data from the 1001 *Arabidopsis* Project [23, 24]. This work aims to add to this body of work by statistically comparing performance between various different machine learning algorithms, including neural networks, using the same dataset.

Importantly, a major concern in the development of machine learning models is overfitting of the data during training and testing, with lower performance being observed on independent datasets. Methods for determining model robustness should therefore be a key priority for the field.

There exist multiple machine learning methods that can be used for supervised prediction tasks such as genomic prediction. Support vector machines, neural networks, random forests, LASSO regression and ridge regression were chosen for this analysis due to their general popularity and ease of implementation. Baseline gBLUP models were also developed in order to gauge the accuracy and robustness of the prediction models.

## Performance

The coefficient of determination is sometimes used in measuring the performance of a genomic prediction model. However, the interpretation of $R^2$ as the amount of variance explained by the model does not hold for non-linear models, and caution is warranted when employing this metric on such models [25–27].

This is unfortunate, as the $R^2$ lies on the same scale as the heritability and would be useful in benchmarking performance against the maximum prediction theoretically possible through

using genomic information. Instead, an alternative metric that is often employed in plant and animal genetics is Pearson's correlation coefficient ($\rho$) between predictions and measurements [28].

In order to estimate predictive performance on outside datasets, a hold-out set or a cross-validation strategy is usually applied to machine learning procedures [15]. *k*-fold cross-validation is generally the preferred method of choice, and involves splitting the data *k* times, each time resulting in an independent training and validation set of individuals. This way, the model is trained and evaluated *k* times, and gives a more accurate estimate of the prediction capability on independent test data. Overfitting is avoided, as would happen in the naive approach whereby a model is trained and then evaluated on the entire dataset. An additional benefit is that all of the data are utilized for training purposes, unlike the hold-out method, resulting in greater power. In the case where hyperparameters are also being selected, a nested cross-validation procedure is required in order to avoid information leak (and thus an overly-optimistic result) from the validation sets [29–31]. The nested approach involves further performing an *n*-fold cross-validation on each of the *k* training sets, resulting in inner training and validation sets alongside the original outer sets (see S1 Fig for a stepwise explanation of the procedure). The best hyperparameter combinations are chosen within each inner *n* split and then applied independently on each outer training set before being evaluated on the final *k* outer validation sets.

In the non-nested approach, one would end up choosing the best hyperparameter values from repeated testing on the *k* validation sets, resulting in an optimistic estimate of the actual predictive performance. When the cross-validation is nested, the hyperparameter values are only configured within the inner *n* loops and so the error when applied to the outer validation sets should result in a more realistic estimate of the performance on an independent dataset.

This is not commonly implemented in the genomic prediction literature, despite the risks that simple cross-validation entails [32]. Thus, in this paper we chose to employ a fully nested cross-validation hyperparameter search in the absence of an available independent dataset. The resulting estimated performance should be relatively free from bias introduced by both hyperparameter tuning and model training.

## Materials and methods

### Data

Open-access *Arabidopsis* SNP and corresponding phenotype data was accessed from the 1001 *Arabidopsis* Project and *AraPheno* database [23, 33]. This dense dataset contains 10,707,430 SNPs across all five chromosomes. Individuals are highly inbred resulting in homozygosity at every position in the genome.

"Time to first flowering" was chosen as the main trait of interest as the genetics of the trait has been extensively studied and had the largest sample size of both genotype and phenotype information [23, 34, 35]. This trait was studied across two laboratory temperatures: 10 C and 16 C. The other traits examined are "seed yield" (dried weight in grams) and "leaf area" (in centimeters squared), chosen with respect to sample size. All individuals were grown under standard laboratory procedures and conditions, constraining the environmental component of trait variation.

### Heritability estimation

GREML analysis was used to estimate the additive SNP-based heritability ($h^2_{\mathrm{SNP}}$) for each trait using GCTA software version 1.92.0 [14, 36].

The GRM used for this analysis was built using a set of 585,375 pruned SNPs. The pruning procedure was conducted in PLINK2 and used a 250kb window, 5 bp step-size and an $r^2$ threshold of 0.05 [37]. A minimum minor allele count of 6 was chosen for SNP inclusion.

## Experimental approach

Using genomic data from the samples of *Arabidopsis*, the objective of this analysis was to compare the relative performance of the various machine learning methods to the baseline linear models. As well as gBLUP, we assessed the performance of least absolute shrinkage and selection operator (LASSO) regression, ridge regression, random forests, support vector machines (linear and non-linear), feed-forward neural networks and convolutional neural networks.

The influence of feature set size and effect of prior linear feature selection were also investigated in this analysis. Prediction accuracies of machine learning methods were compared against gBLUP prediction as a baseline, and for each trait the SNP-based heritability was also calculated to gauge the overall performance of the models.

## Feature selection

Two feature selection strategies were adopted in order to select the input SNP set from the dense genotype dataset, with feature selection performed independently on all inner and outer loops of this analysis. The first strategy was to use a GWAS to identify the SNPs most linearly associated with the trait. The results were subsequently clumped to generate the set of top SNPs free from strong linkage disequilibrium (LD).

GWASs were performed on the various *Arabidopsis* phenotypes using PLINK2 GLM association [38]. A minimum allele count of 20 was used, as is recommended for the `--grm` flag, and the resulting SNPs were clumped using the following parameters: significance threshold of 0.05, secondary significance threshold of 0.1, LD threshold of 0.05 and distance threshold of 250kb.

The second approach was to omit an initial GWAS and select SNPs randomly from across the genome. SNPs were first pruned down to an LD-independent set using a 250kb window, 5bp step-size and an $r^2$ threshold of 0.05. A minimum allele count of 6 was used.

## Model optimization

In order to implement nested-cross validation a modified version of the Python NestedCV package was used (https://github.com/casperbh96/Nested-Cross-Validation), allowing for logging and plotting of inner loop test results and pre-specified random cross-validation splits. Cross-validation splits were made with pre-shuffling using Scikit-learn's KFold function [39].

gBLUP prediction was performed in GCTA using the previously described GRMs and the subset of feature selected SNPs [14, 40].

All regression, tree and kernel methods were performed using the Scikit-learn library in Python3. Neural networks were implemented using Keras on a TensorFlow backend [41, 42]. The mean absolute error loss metric was chosen for all machine learning algorithms to prioritize robustness [15].

Genotypes were loaded in as binary variables along with corresponding phenotypes. Phenotype values were standardized and scaled prior to learningusing Scikit-learn's *StandardScaler* function. Fitting was performed on the training set only, with the same transformation applied to both training and testing sets.

Random grid searching was employed to optimize the hyperparameters within the inner loops. The performance of each hyperparameter value across several runs was recorded and

performance box-plots were manually evaluated. Appropriate adjustment of the search-space for subsequent inner loop grid-searching was then implemented.

## Performance assessment

In order to compare predictivity across all methods used, Pearson's correlation coefficient on the test set was calculated.

A multiple-comparison Dunnett test was used in order to test for significant differences in performance between the machine learning models and the baseline gBLUP model [43, 44]. This was implemented in a non-parametric and small sample size design using the nparcomp and mlt packages in R-3.6.3 [45–47].

## Results

### Heritability estimation

The SNP-based narrow-sense heritabilities for each trait are shown in Table 1. For the flowering time traits, the values are relatively high, which may be explained by the fact that non-genetic variance was constrained by the laboratory conditions under which the plants grew. Given that nearly all variation in flowering time is captured by genetic effects, it is likely that genomic prediction models could be useful in accounting for trait variance in the population.

For the other two traits, seed yield and leaf area, the heritabilities were lower. It is possible that their smaller sample size contributed to this result, as well as a larger element of random or developmental variation underlying the traits.

### Genomic prediction

The results for the flowering traits can be seen in Figs 1 and 2. Some machine learning families consistently outperform the gBLUP method, often significantly so. Importantly, this was not true for all families and shows that the linear gBLUP serves as as important baseline upon which to improve. For context in terms of heritability and of the amount of variation being explained by the models, the $R^2$ of the gBLUP models are given in the figure legends.

Feed-forward neural networks were the best method on average in terms of overall ranking, as well as the model that was significantly improved from the baseline the most times. For this reason, they could be seen as the machine learning family that has the most potential to improve over gBLUP. Although neural networks tend to have a lot of tuned hyperparameters, and are thus prone to overfitting, this was not seen to a detrimental extent in the outer loops of the nested cross-validation. Ridge regression, which is a linear model, also performed generally quite well.

Random forests were seen to be the most predictive model in some of the sets, but also often failed to improve over the gBLUP, making it hard to strongly recommend. Convolutional neural networks also tended to have good predictive performance except in the cases where there was clear failure on one of the cross-validation sets due to model misspecification

**Table 1. SNP-based narrow-sense trait heritabilities ($h^2_{\text{SNP}}$) of four *Arabidopsis thaliana* traits.**

| Phenotype | N | $h^2_{\text{SNP}}$ | s.e. |
|---|---|---|---|
| Flowering Time (10 C) | 1058 | 0.943 | 0.057 |
| Flowering Time (16 C) | 1021 | 0.904 | 0.057 |
| Seed Yield | 384 | 0.236 | 0.101 |
| Leaf Area | 445 | 0.300 | 0.093 |

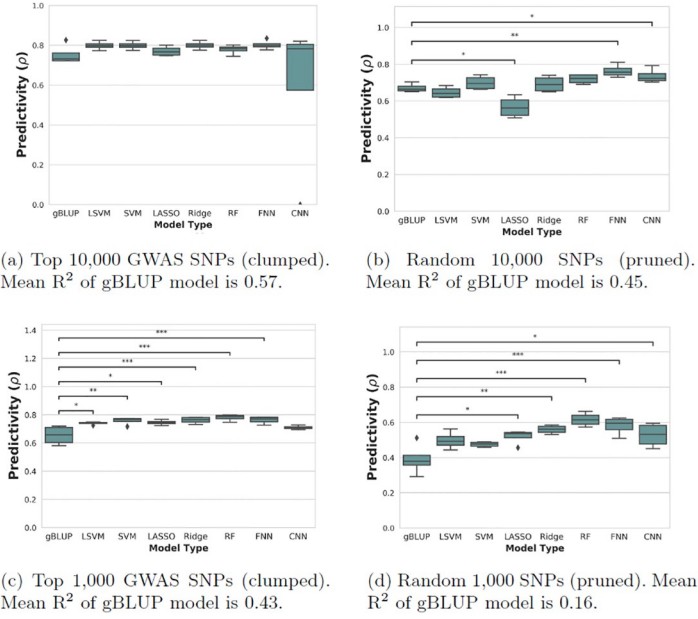

(a) Top 10,000 GWAS SNPs (clumped). Mean $R^2$ of gBLUP model is 0.57.

(b) Random 10,000 SNPs (pruned). Mean $R^2$ of gBLUP model is 0.45.

(c) Top 1,000 GWAS SNPs (clumped). Mean $R^2$ of gBLUP model is 0.43.

(d) Random 1,000 SNPs (pruned). Mean $R^2$ of gBLUP model is 0.16.

**Fig 1. Nested cross-validation results for flowering time (10 C).** Each box plot represents k = 4 outer validation sets. Dunnett test p-value annotations: *, $p < 0.05$; **, $p < 0.01$; ***, $p < 0.001$. gBLUP, genomic best linear unbiased prediction; CNN, convolutional neural network; RF, random forests; LASSO, least absolute shrinkage and selection operator regression; SVM, support vector machine; LSVM, linear SVM; Ridge, ridge regression; FNN, feed-forward neural network.

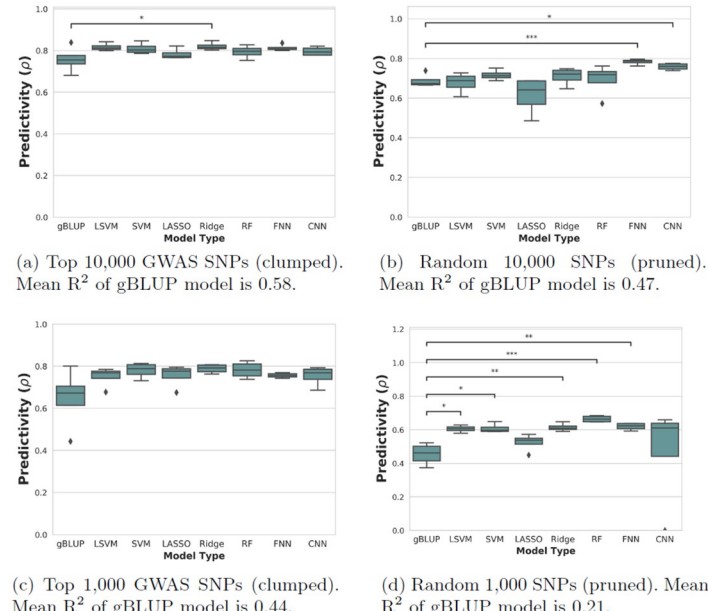

(a) Top 10,000 GWAS SNPs (clumped). Mean $R^2$ of gBLUP model is 0.58.

(b) Random 10,000 SNPs (pruned). Mean $R^2$ of gBLUP model is 0.47.

(c) Top 1,000 GWAS SNPs (clumped). Mean $R^2$ of gBLUP model is 0.44.

(d) Random 1,000 SNPs (pruned). Mean $R^2$ of gBLUP model is 0.21.

**Fig 2. Nested cross-validation results for flowering time (16 C).** Each box plot represents k = 4 outer validation sets. Dunnett test p-value annotations: *, $p < 0.05$; **, $p < 0.01$; ***, $p < 0.001$. gBLUP, genomic best linear unbiased prediction; CNN, convolutional neural network; RF, random forests; LASSO, least absolute shrinkage and selection operator regression; SVM, support vector machine; LSVM, linear SVM; Ridge, ridge regression; FNN, feed-forward neural network.

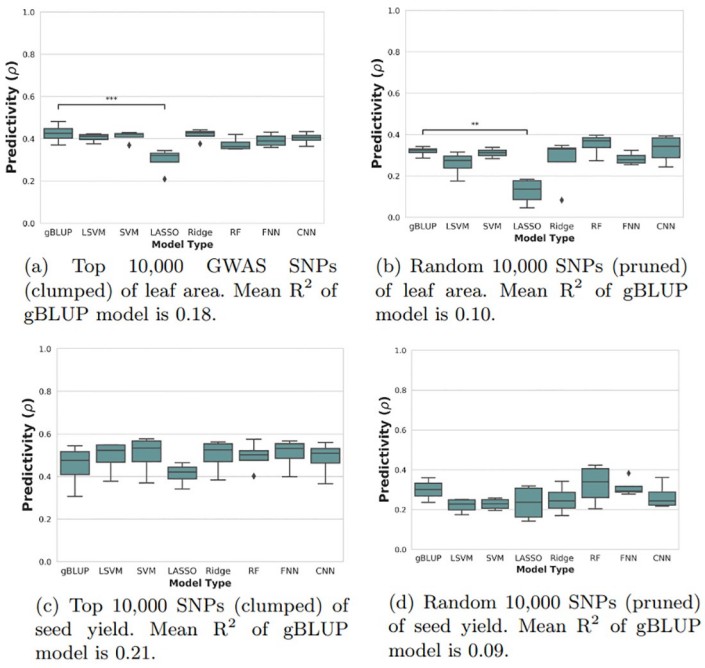

(a) Top 10,000 GWAS SNPs (clumped) of leaf area. Mean $R^2$ of gBLUP model is 0.18.

(b) Random 10,000 SNPs (pruned) of leaf area. Mean $R^2$ of gBLUP model is 0.10.

(c) Top 10,000 SNPs (clumped) of seed yield. Mean $R^2$ of gBLUP model is 0.21.

(d) Random 10,000 SNPs (pruned) of seed yield. Mean $R^2$ of gBLUP model is 0.09.

**Fig 3. Nested Cross-Validation Results for seed yield and leaf area.** Each box plot represents k = 4 outer validation sets. Dunnett test p-value annotations: *, $p < 0.05$; **, $p < 0.01$; ***, $p < 0.001$. gBLUP, genomic best linear unbiased prediction; CNN, convolutional neural network; RF, random forests; LASSO, least absolute shrinkage and selection operator regression; SVM, support vector machine; LSVM, linear SVM; Ridge, ridge regression; FNN, feed-forward neural network.

(see Figs 1a and 2d). This instability in prediction is an important observation and would not have been seen if not for the nested design. For this reason, in future model-building, it would be recommended to look closely for full robustness on any validation sets before moving to a final test set.

Two overall trends are also clear from the plots: that using a larger SNP-set size leads to improved performance; and that feature selection through linear association (GWAS) also aids in decreasing predictive error.

The statistical comparisons between models for the other two traits, seed yield and leaf area, can be seen in Fig 3. Unlike with the flowering traits, none of the machine learning models was able to significantly outperform the baseline gBLUP method. In fact, some of the models significantly underperformed, particularly LASSO regression.

## Discussion

The results presented here show the potential for non-linear models to improve upon linear genomic prediction in *Arabidopsis*. The possibility for substantial improvement is worth pursuing given the struggles in the field to close the missing heritability gap despite large sample sizes and dense marker sets. Improvements in prediction were dependant on model type, feature set size, feature selection method, and trait.

Overall, the baseline gBLUP model yielded reasonable predictions in comparison to the machine learning models, a not unexpected result from prior research. However, for flowering traits, some machine learning families could be seen to perform consistently better than the baseline model, such as neural networks and ridge regression. The predictive performance of neural networks is in line with similar work from Raimondi et al. [24].

This improvement in performance was often significant, despite the lower statistical power the cross-validation design gave for performance assessment. Several previous studies on machine learning in genomic prediction have omitted statistical tests, presumably on the basis of low power, although it is an important component of assessing improvements in performance.

For the other two traits, none of the machine learning models was able to significantly out-perform the baseline model. These traits had lower heritabilities and smaller sample sizes than flowering time which could be contributing factors to this result. This adds to the existing body of evidence that improvement in performance using machine learning is not guaranteed and is highly dependant on trait and dataset.

Performance in prediction generally improved when using more SNP markers. Despite a much smaller number of individuals than predictors, overfitting with so many variables did not seem to be an issue, as might happen in $P \gg N$ problems.

Additionally, prior linear association for feature selection was found to be a useful step in this study, which is not guaranteed when modelling non-additive interactions [48, 49]; this performance boost was not extreme, however.

Interestingly, the performance improvement for the best non-linear models over the linear models was more pronounced when using random SNPs, which, unlike the top GWAS SNPs, were not chosen for their prior linear association. It is perhaps understandable that the most linearly associated features tend to explain the most variance when combined in a linear fash-ion; however, it is evident that a substantial amount of variation stands to be captured from the rest of the genome, where non-linear effects may play a larger role [50, 51]. For this reason, it is often noted, and this study reiterates, that linear feature selection may overlook important variables that combine non-linearly.

Notwithstanding this possibility, due to computational limitations, a subset of SNPs had to be chosen as input in this study and it was found from this study that the optimal input fea-tures were the most significant QTLs obtained from a GWAS. As compute power grows, how-ever, one should remain open-minded as to the input search space when using genotypic information to predict complex traits.

For example, it is possible that widespread biological and statistical epistasis exists for these traits which is why the non-linear models were sometimes successful in explaining more trait heritability. It is unclear the extent to which this might be the case in this work. A better under-standing of the total genetic architecture of a trait could aid in building more powerful explan-atory models [9]. One of the drawbacks of machine learning, however, is that the models are often "black boxes" that are not readily interpretable [15]. Although improved prediction can hint towards a greater role for epistasis in the variance of complex traits, elucidating the spe-cific complex interactions is not a simple task. One possible extension of this study would be to find putative epistatic interactions through linear methods in the feature selection step.

Genomic prediction is concerned not just with performance, but also with ensuring that methods and models are robust enough to be reliable, consistent and transferable across data-sets. This study used a nested-cross validation approach for hyperparameter tuning in order to assure the reliability of the results. This approach was useful in identifying a pitfall of hyper-parameter tuning using $k$-fold cross-validation, namely the potential to miss serious misspeci-fication on the final models despite success on validation sets.

Limitations of this analysis include the modest sample sizes for the various traits and the lack of a fully independent external test dataset. In other datasets collected from breeding field populations, there would also be a larger non-genetic component of variation, as well as geno-type-environment interactions, which could possibly affect the relative performances of the various machine learning models. Furthermore, the individuals in this analysis were inbred to

homozygosity which simplified the input genotype matrices and removed intra-allelic effects such as dominance. Any of these considerations could cause a difference in model performance in natural populations [52].

As the feature selection step was linear in this analysis, the GRM employed in the MLMA should have somewhat mitigated the effect of population structure, yet the issue of confounding is a central one in genomic prediction and it not easily addressed here.

Firstly, in the field of machine learning, the most appropriate methods for handling confounding variables which are often needed in genetic studies, such as through using the first principal components of variation as linear covariates in a regression, have not been well established [53–55].

Secondly, the individuals in this analysis were grown under standardized laboratory conditions which theoretically removes the effect of confounding by means of a correlation between genotype and environmental exposure (e.g. differences in soil type and mineral concentrations). However, to the extent that there is correlation between ancestry informative SNPs and causal loci (and ignoring genotype-environment interactions), fully accounting for this population structure may reduce predictive performance in such a way as to be undesirable [56, 57]. As the final model would ultimately be predicting phenotype from genotype, such information should not necessarily be excluded when trying to maximize the explained variance across real populations in potentially varied environments.

Thirdly, although many machine learning models are black boxes in terms of causal inference, it might still be an important goal of a prediction tool to identify relevant genes, variants and pathways that are being used by the model.

For the above stated reasons, we have left ascertaining the best approach to handling confounding and covariate adjustment in using machine learning for genomic prediction to future work on separate datasets with known environmental confounders.

## Supporting information

**S1 Fig. A nested cross-validation procedure.**
(TIF)

**S1 File. Grid search parameters.**
(DOCX)

## Acknowledgments

Many thanks to Ross Byrne, Mark Doherty, Jennifer Hengeveld, Laura O'Briain, the Bradley lab and the Cassidy lab.

## Author Contributions

**Conceptualization:** Ciaran Michael Kelly, Russell Lewis McLaughlin.

**Formal analysis:** Ciaran Michael Kelly.

**Funding acquisition:** Russell Lewis McLaughlin.

**Investigation:** Ciaran Michael Kelly.

**Methodology:** Ciaran Michael Kelly, Russell Lewis McLaughlin.

**Supervision:** Russell Lewis McLaughlin.

**Validation:** Ciaran Michael Kelly.

**Visualization:** Ciaran Michael Kelly.

**Writing – original draft:** Ciaran Michael Kelly.

**Writing – review & editing:** Russell Lewis McLaughlin.

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
