## [Decision Letter · Decision Letter 0]

12 Feb 2024

PONE-D-23-39918Comparison of machine learning methods for genomic prediction of Arabidopsis thaliana traitsPLOS ONE

Dear Dr. Kelly,

Thank you for submitting your manuscript to PLOS ONE. After careful consideration, we feel that it has merit but does not fully meet PLOS ONE’s publication criteria as it currently stands. Therefore, we invite you to submit a revised version of the manuscript that addresses the points raised during the review process.

We look forward to receiving your revised manuscript.

Kind regards,

Lewis Lukens

Academic Editor

PLOS ONE

Journal Requirements:

"This study was supported by Science Foundation Ireland (17/CDA/4737). Russell McLaughlin also receives support from the MND Association (879-791) and Science Foundation Ireland

(16/RC/3948)."

4. Please expand the acronym “MND” (as indicated in your financial disclosure) so that it states the name of your funders in full.

5. Please note that in order to use the direct billing option the corresponding author must be affiliated with the chosen institute. Please either amend your manuscript to change the affiliation or corresponding author, or email us at plosone@plos.org with a request to remove this option.

6. Please upload a copy of Supporting Information Figure/Table/etc. "S1 Grid Search Parameters" which you refer to in your text on page 10/14.

**Additional Editor Comments:**

Thank you for your submission. We received two reviews of your work, and I also reviewed the article. Please address all of the reviewers’ concerns.

A number of reviewers’ concerns were shared. Both reviewers would like to see the code and data set made available. Both reviewers were concerned that the article does not sufficiently explain its contribution to science. Finally, both reviewers were concerned with model accuracy and how population structure affected it.

My points also overlap with these concerns. First, the article’s premise should be clarified. The idea seems to be that additive linear models do not capture non-additive interactions among loci, so machine learning methods should be applied for prediction. However, linear models do capture epistasis. As one empirical example, we have shown that linear model analysis of Arabidopsis progeny whose traits are fully explained by two-gene epistasis will nonetheless assign additive effects to parental lines. Since additive models capture variance due to epistasis, the models may predict genotypic values quite well, but the genotypic values may not predict offspring values well (e.g. see line 11). A second issue is that the manuscript can’t ascribe performance differences between machine learning methods and linear models to ML abilities to capture epistasis. Methods predictions vary for unknown reasons. Some ML methods have better predictive abilities than additive models and some worse. I would remove text from the abstract, introduction, and elsewhere that attributes differences in models’ performances to known genetic phenomena (e.g. paragraphs 2-4).

Second, I am not very familiar with the 1001 genome project, but given the lines are from different latitudes and altitudes, population structure is causing the marker-flowering time associations. As a result, the paper’s objective seems to be to evaluate methods that best capture population differences. This does not seem useful. If one can predict a genotype’s flowering time based on latitude and altitude, why would one perform genomic predictions? It would be more interesting to predict traits after removing population structure. Even removing the effects of each entry’s latitude and altitude collection site may have a major effect on results.

Finally, predictions would likely be improved by using more SNPs. Why were 1,000 or 10,000 SNPs used? This seems very low, and I suspect many more could be used without running into “computational limitations.” To look into this, I’d plot the number of markers and their predictions for different methods. Even if there are computational limitations with some approaches and not others, it is still worth knowing the best predictions you can get with the all data at hand.

I found a few sentences and topics confusing. For example, the lines were grown in one experimental condition. Because there is no G x E, I did not understand why it is discussed at the end of the manuscript.

Reviewers' comments:

Reviewer's Responses to Questions

**Comments to the Author**

1. Is the manuscript technically sound, and do the data support the conclusions?

Reviewer #1: Partly

Reviewer #2: Partly

2. Has the statistical analysis been performed appropriately and rigorously? 

Reviewer #1: Yes

Reviewer #2: Yes

3. Have the authors made all data underlying the findings in their manuscript fully available?

Reviewer #1: No

Reviewer #2: No

4. Is the manuscript presented in an intelligible fashion and written in standard English?

Reviewer #1: Yes

Reviewer #2: Yes

5. Review Comments to the Author

Reviewer #1: The manuscript by Kelly and McLaughlin presents an analysis of using various Machine Learning (ML) algorithms to perform genomic prediction, and compares the performance of these methods with the more traditional gBLUP approach. Such comparison could be quite useful, but recently various authors have presented similar results which are not cited in the manuscript (e.g. https://www.ncbi.nlm.nih.gov/pmc/articles/PMC10080209/;
https://pubmed.ncbi.nlm.nih.gov/34792168;
https://www.frontiersin.org/articles/10.3389/fpls.2022.932512/full; and various others). It would be good to describe more comprehensively what was already learned in these existing publications.

This leads to my major concern:

(1) What does this manuscript add to the results presented in Raimondi et al. (2022) [https://pubmed.ncbi.nlm.nih.gov/34792168/]. In that paper, much more traits are used from the same Arabidopsis dataset as used in the current manuscript; it is also shown, similar as in the current manuscript, that neural networks can do well; and it is also shown, similar as in the current manuscript, that flowering time is one of the type of traits which are predicted best. It would be important to refer to this study and also make more clear what is done differently in the current work in order to obtain useful additional results, and to describe what insights are obtained that are not available in Raimondi et al.

(2) In any case, it is unclear why the four specific traits which are used in the manuscript, were chosen out of the larger set of traits available in the Arabidopsis dataset which is used. Some argumentation for this would be needed, or better, a more extensive analysis of the different traits would be warranted.

(3) Code used for training the machine learning models does not seem available.

In addition, some minor comments:

(4) How about relatedness between the individuals - how does this influence prediction performance? Should this not be taken into account in making training-test splits (to prevent that out of two very related individuals, one is in the training set and the other in the test set)?

(5) Two sentences in the methods could be further clarified (p.6): “Phenotype values were standardized and scaled based on the training set prior to learning"; and "Random grid searching and manual evaluation on the inner loops was implemented to optimize the hyperparameters". It would be good to give additional details here on how this was done.

Reviewer #2: Major Comments:

1- The data is not provided in the github repository and therefore, scripts are not working. Moreover, I could not find scripts of ML methods.

2- Population structure and data specifics are not discussed. Also, population structure is not incorporated (though discussed), so the predictive abilities can be overestimated. I would suggest incorporating it in any case for comparison or show the specifics of the genotype data, if it is not significant.

3- GBLUP and Ridge are showing different performances. However, literature suggests their equivalence (Meuwissen et al. 2001).

4- I would appreciate to see the outcomes with full genotype dataset.

5- Pruning using linear methods restricts it to the SNPs with main effects. This would remove any SNPs with interactions and having minor main effects. For this purpose, I would suggest including some non-linear feature selector.

6- For a high heritability, we can expect the model to make minimal errors and the accuracy should achieve theoretical maximum. In your case, it saturates to ~0.8, which is slightly lower. Would be please explain it, why?

Minor Comments:

1- I would like to see why 1001 genome project’s population would be a good to test GP models and how would it be consistent with a more realistic breeding population.

2- In the abstract and lines 12-14, it was mentioned that linear models may be impeded by their inability to make use of non-additive effects. This may not be true because many linear models can accommodate non-additive effects. For instance, EGBLUP by Martini et al., 2017

3- In the abstract, it was mentioned that developing complex models with larger SNP sets can improve performance. I don't think it is right, because complex models tend to be overfitted as well.

4- The introduction section doesn't connect this work to some earlier studies on GP using A. thaliana very well. For instance, a recent work by Farooq et al., 2022 etc. on this topic.

5- Only three traits were explored but the conclusions and the title/abstract seem overambitious. May be narrowing down the scope to these three traits and explicitly mentioning this in the title / abstract would help the reader.

6- Line 42 says GBLUP uses fixed effects for SNPs, whereas it uses random effects and solves SNP effects through mixed model’s equations.

7- Line 69-70 says that all ML methods are non-linear, whereas Ridge and Lasso are linear.

8- Lines 90-92: please explain why only these methods were chosen.

9- I don't think figure 1 adds much information to the main text. It can be in the supplementary.

10- Line 136: A. thaliana has 5 chromosomes, whereas you mentioned it as four.

11- Line 205: you mentioned that non-genetic variance was constrained by the laboratory conditions. Yes, that can be the case, but we should still expect epistasis and dominance, though, GxE is limited.

6. PLOS authors have the option to publish the peer review history of their article (what does this mean?). If published, this will include your full peer review and any attached files.

Reviewer #1: No

Reviewer #2: **Yes: **Muhammad Farooq (PhD)

---

## [Author Response · Author response to Decision Letter 0]

5 Apr 2024

Please see uploaded response to reviewers document as answers are given to each individual point. Thank you for your time.

---

## [Decision Letter · Decision Letter 1]

5 Jun 2024

PONE-D-23-39918R1Comparison of machine learning methods for genomic prediction of selected Arabidopsis thaliana traitsPLOS ONE

Dear Dr. McLaughlin,

Thank you for submitting your manuscript to PLOS ONE. After careful consideration, we feel that it has merit but does not fully meet PLOS ONE’s publication criteria as it currently stands. Therefore, we invite you to submit a revised version of the manuscript that addresses the points raised during the review process by Reviewer 2.

We look forward to receiving your revised manuscript.

Kind regards,

Lewis Lukens

Academic Editor

PLOS ONE

Journal Requirements:

Reviewers' comments:

Reviewer's Responses to Questions

**Comments to the Author**

1. If the authors have adequately addressed your comments raised in a previous round of review and you feel that this manuscript is now acceptable for publication, you may indicate that here to bypass the “Comments to the Author” section, enter your conflict of interest statement in the “Confidential to Editor” section, and submit your "Accept" recommendation.

Reviewer #1: All comments have been addressed

Reviewer #2: (No Response)

2. Is the manuscript technically sound, and do the data support the conclusions?

Reviewer #1: Yes

Reviewer #2: Yes

3. Has the statistical analysis been performed appropriately and rigorously? 

Reviewer #1: Yes

Reviewer #2: Yes

4. Have the authors made all data underlying the findings in their manuscript fully available?

Reviewer #1: Yes

Reviewer #2: No

5. Is the manuscript presented in an intelligible fashion and written in standard English?

Reviewer #1: Yes

Reviewer #2: Yes

6. Review Comments to the Author

**Reviewer #1: **(No Response)

**Reviewer #2: **1- The data is still not provided in the GitHub repository. Instead, it referred to the 1001 genome project page, which doesn’t allow it download straightforward. I would suggest it to be there in the git or the code should allow to reproduce the results by fetching it inside. The source code is still not fully reproducible. The article should not be online without having it.

2- Line 166-167: “Time to first flowering was chosen as the main trait of interest as the genetics of the trait has been extensively studied”. Please cite some references to show if it is well studied.

3- Line 198 and 203: different minimum allele counts were used. Please state why? Also, don’t you think it will create a bias for the comparison?

4- Predictions on all data are still missing (see last comments).

5- The order of model names should be consistent between figures on the horizontal axis for better readability and comparison. Also, the font sizes must be increased. Moreover, please mention how many data points of accuracies each boxplot represents. I have already suggested using multiple repeats of nested cross-validations for tackling PS.

7. PLOS authors have the option to publish the peer review history of their article (what does this mean?). If published, this will include your full peer review and any attached files.

Reviewer #1: No

Reviewer #2: **Yes: **Muhammad Farooq

---

## [Author Response · Author response to Decision Letter 1]

19 Jul 2024

Points still need to be addressed:

1- The data is still not provided in the GitHub repository. Instead, it referred to the 1001 genome project page, which doesn’t allow it download straightforward. I would suggest it to be there in the git or the code should allow to reproduce the results by fetching it inside. The source code is still not fully reproducible. The article should not be online without having it.

Response: The links to all genotype and phenotype downloads have now been given in the GitHub repository and the README has been updated to reflect this. We thank you for this comment. 

2- Line 166-167: “Time to first flowering was chosen as the main trait of interest as the genetics of the trait has been extensively studied”. Please cite some references to show if it is well studied.

Response: This is a fair point. We have now given references an interested can reader can follow-up on including a classic early paper exploring flowering time as a quantitative trait. 

3- Line 198 and 203: different minimum allele counts were used. Please state why? Also, don’t you think it will create a bias for the comparison?

Response: This was an omission on our part and thank the reviewer for the observation. The PLINK docs recommend a minimum MAC filter of 20 when using the GLM. “The statistics computed by --glm are not calibrated well when the minor allele count is very small. --mac 20 is a reasonable filter to apply” (see https://www.cog-genomics.org/plink/2.0/assoc). To the extent that it may or may not affect the results, it is a limitation of the statistical analysis used when employing PLINKS GLM that is not present in the other approach. 

4- Predictions on all data are still missing (see last comments).

Response: All predictions are now available on the GitHub repository at https://github.com/ciaranoceallaigh96/arabidopsis_ml/blob/main/ml_predictions.zip and https://github.com/ciaranoceallaigh96/arabidopsis_ml/blob/main/pheno_and_gblup_predictions.zip and the README has been updated to reflect this. 

5- The order of model names should be consistent between figures on the horizontal axis for better readability and comparison. Also, the font sizes must be increased. Moreover, please mention how many data points of accuracies each boxplot represents. I have already suggested using multiple repeats of nested cross-validations for tackling PS.

Response: As requested, we have changed the order of the model names and increased the font size as much as possible. We have also updated the figure legend to be clearer on the number of data points.

---

## [Editor Report · Decision Letter 2]

5 Aug 2024

Comparison of machine learning methods for genomic prediction of selected Arabidopsis thaliana traits

PONE-D-23-39918R2

Dear Dr. McLaughlin,

Thank you for the manuscript revision. We’re pleased to inform you that your manuscript has been judged scientifically suitable for publication and will be formally accepted for publication once it meets all outstanding technical requirements.

Kind regards,

Lewis Lukens

Academic Editor

PLOS ONE

---

## [Editor Report · Acceptance letter]

15 Aug 2024

PONE-D-23-39918R2 

PLOS ONE

Dear Dr. McLaughlin, 

I'm pleased to inform you that your manuscript has been deemed suitable for publication in PLOS ONE. Congratulations! Your manuscript is now being handed over to our production team.

Kind regards, 

on behalf of

Dr. Lewis Lukens 

Academic Editor

PLOS ONE